# Induction of Hibernation and Changes in Physiological and Metabolic Indices in *Pelodiscus sinensis*

**DOI:** 10.3390/biology12050720

**Published:** 2023-05-15

**Authors:** Runlan Lin, Jiahao Wu, Ziyi You, Dongjie Xu, Caiyan Li, Wei Wang, Guoying Qian

**Affiliations:** 1College of Fisheries and Life Science, Shanghai Ocean University, Shanghai 201306, China; 2College of Biology and Environment, Zhejiang Wanli University, Ningbo 315100, China

**Keywords:** *P. sinensis*, hibernation, metabolism, histones, methylation

## Abstract

**Simple Summary:**

Hibernating animals reduce their metabolic rate through a variety of transcriptional, translational, and post-translational regulation methods to survive in extreme environments with an overall reduction in their body’s activity levels. Histone methylation is a mechanism that regulates gene expression at the transcriptional level, and specific histone methylation and demethylation modifications can either inactivate or activate gene transcription. In this study, we focused on the hibernation of *Pelodiscus sinensis* (*P. sinensis*) as an experimental model and studied the changes in the physiological metabolism index, histones, and related genes on growth and development under conditions of artificially induced hibernation. Although this paper lacks further evidence that *P. sinensis* has been induced into a deep hibernation state, cold torpor is perhaps a more accurate description. The changes in physiological indexes, transcription level, mRNA relative expression, protein localization, relative protein expression, and enzyme activity associated with histone and histone (demethylation) genes in various tissues were analyzed under hibernating and normal conditions. The results provide insight into the adaptative mechanism of *P. sinensis* to its environment.

**Abstract:**

*Pelodiscus sinensis* (*P. sinensis*) is a commonly cultivated turtle species with a habit of hibernation. To study the changes in histone expression and methylation of *P. sinensis* during hibernation induction, a model was established by artificial induction. Physiological and metabolic indices were measured, and the expression and localization of histone (*H1*, *H2A*, *H2B*, *H3*, and *H4*) and methylation-related genes (*ASH2L*, *KMT2A*, *KMT2E*, *KDM1A*, *KDM1B*, and *KDM5A*) were measured by quantitative PCR, immunohistochemistry, and Western blot analysis. The results indicated that the metabolism, antioxidation index, and relative expression of histone methyltransferase were significantly decreased (*p* < 0.05), whereas the activity and expression of histone demethyltransferase were significantly increased (*p* < 0.05). Although our results showed significant changes in physiological and gene expression after hibernation induction, we could not confirm that *P. sinensis* entered deep hibernation. Therefore, for the state after cooling-induced hibernation, cold torpor might be a more accurate description. The results indicate that the *P. sinensis* can enter cold torpor through artificial induction, and the expression of histones may promote gene transcription. Unlike histones expressed under normal conditions, histone methylation may activate gene transcription during hibernation induction. Western blot analysis revealed that the *ASH2L* and *KDM5A* proteins were differentially expressed in the testis at different months (*p* < 0.05), which may perform a role in regulating gene transcription. The immunohistochemical localization of *ASH2L* and *KDM5A* in spermatogonia and spermatozoa suggests that *ASH2L* and *KDM5A* may perform a role in mitosis and meiosis. In conclusion, this study is the first to report changes in histone-related genes in reptiles, which provides insight for further studies on the physiological metabolism and histone methylation regulation of *P. sinensis* during the hibernation induction and hibernation period.

## 1. Introduction

*Pelodiscus sinensis* (*P. sinensis*) is a hibernating reptile that can tolerate extreme environments of low temperature and low oxygen. The hibernation behavior of animals may also be achieved by artificial induction. During hibernation, the heart rate, oxygen consumption, and metabolism of animals are reduced [1,2,3,4,5]. *Chrysemys picta* [6], *Spermophilus tridecemlineatus* [7,8], wild-type mice [9], and *Ursus americanus* [10] have been subjected to artificially induced hibernation, and their body temperature, heart rate, and pyruvate dehydrogenase and creatine kinase content were reduced. During hibernation, the liver of *Ictidomys tridecemlineatus* [11] and the heart and muscle of grizzly bears [12] exhibit significantly decreased transcription levels and metabolic rates, and DNA and protein synthesis is also inhibited [13,14]. Metabolic inhibition results in decreased DNA synthesis and histone expression. Histone ends are methylated, and gene expression patterns are regulated by changes in chromatin structure and DNA [15,16]. In particular, histone H3 lysine 4 methylation (*H3K4me*) is associated with gene-specific activation and development and is dynamically regulated by histone methylation and demethylation [17]. Histone methylation (HMT) performs a role in transcriptional activation and includes Absent, small, or homeotic 2-like (*ASH2L*) and histone demethylation (DM) 2A and 2E (*KMT2A*/*MLL1*, *KMT2E*/*MLL5*). The corresponding histone demethylation genes primarily include histone demethylation 1A, 1B, and 5A (*KDM1A*/*LSD1*, *KDM1B*/*LSD2*, *KDM5A*/*RBP2*). Studies have shown that deep hibernation is associated with chromatin condensation, which is reversed upon awakening [18]. Histone methylation is also involved in chromatin remodeling and is associated with gene-specific activation and developmental processes. The high expression of *ASH2L*, *KMT2A,* and *KMT5A* is positively associated with the activation of early embryonic development [19], and the lack of the histone demethylation genes *LSD1*, *KDM1A,* and *KDM5A* result in the failure of embryonic development [19].

Whether hibernation causes changes in physiological and biochemical indicators and transcriptional activity in *P. sinensis*, thereby affecting the expression of histones and related genes, is unclear. Therefore, we artificially induced hibernation to study the changes in physiological and metabolic indicators and the histones of *P. sinensis* on growth and development during hibernation. This enables the study of the physiological metabolism of hibernating animals without seasonal restrictions and provides a reference model for the artificially induced hibernation of reptiles. The analysis of physiological indicators and histone gene expression data during hibernation will also provide a reference for histone expression in *P. sinensis*. The results provide further insight into the regulatory mechanism of hibernation in reptiles, such as *P. sinensis*.

## 2. Materials and Methods

### 2.1. Establishment of a Hibernation Induction Model

Healthy *P. sinensis* (one-year-old, 200–250 g, male *n* = 25; female *n* = 25) were purchased in June from the Walmart Wanda Plaza store in Ningbo, Zhejiang Province. The hibernation experiment was carried out in our laboratory. An artificial hibernation model was induced by cooling down and controlling the diet of the animals [7,19,20]. *P. sinensis* was raised in a single cage in a plastic box with a lid, the water level did not exceed the highest point of the carapace, and the box was placed in a constant temperature incubator (BXS-250, Shanghai Boxun Industrial Co. Ltd. (Shanghai, China)) for the 24h-dark light environment [7]. One week later, once *P. sinensis* showed no excrement, temperature reduction was initiated. Starting from 27 °C, the temperature was decreased by 1 °C each day. To ensure water quality, one-third of the water was replaced isothermally every day and completely replaced every three days. The dorsal and plastron temperature was measured and recorded (Infrared thermometer, FLUKE Company (Shanghai, China)), which included the room temperature, water temperature, and oral temperature (measured by mercury thermometer). In order to reduce the influence of the determination on the hibernating state, 5 males and 5 females were selected in turn for the determination. A finger-clip pulse oximeter (Shanghai Haier Medical Technology Co., Ltd. (Shanghai, China)) was used to record the number of heartbeats and blood oxygen concentration within 30 s. The turtle was turned over to calculate the time it took to turn over, and the length of its neck extension was measured with a ruler. Beginning at 20 °C, the temperature was decreased by 3 °C every two days, and the above indicators were measured daily. Once 11 °C was reached, the temperature was decreased by 1 °C every three days, and cooling was stopped at 8 °C.

The ethical standards of the animal experiment scheme have passed the ethical review of the experimental animal ethics committee of Zhejiang Wanli University. The acceptance number of the ethical review is 2020070101.

### 2.2. Sample Collection

The acquisition and use of animals followed the regulations for the management and use of laboratory animals in China. After the hibernation induction experiment, three males and three females were randomly selected as parallel samples, and their body weights were weighed with an electronic balance (accuracy = 0.01 g). Following sedation with an intraperitoneal injection of sodium pentobarbital (20 mg/kg), they were sacrificed by cervical dislocation. The hibernation induction samples were conducted on ice. The heart, liver, spleen, lung, kidney, ovary, muscle, adipose, small intestine, and testis were washed three times with sterile saline, transferred to cryopreservation tubes, and snap frozen in liquid N_2_. All tissues were stored at −80 °C for future use. Part of the testicular tissue was washed with distilled water and stored in 4% paraformaldehyde for 48 h. The solution was replaced with 50% alcohol for 2 h and stored in 70% alcohol for future use.

### 2.3. Determination of Metabolism and Physiological Indicators

The following physiological indicators were measured in the samples.

Determination of glycogen content: 50 mg of liver and muscle tissues were mixed with 1.5 mL of 30% KOH and boiled in a water bath for 15 min. After cooling, the mixture was transferred to a 50 mL volumetric tube, and water was added to the mark and mixed. Glycogen content was determined based on the procedure described by Laurentin [21].

Glycolytic lactate assay: Liver and muscle tissue (2 portions of 50 mg) were transferred to a 2 mL sample tube and a control tube. Then, 300 μL of PBS (0. 01 M, pH 7.4) was added, and the samples were homogenized. The determination of glycolytic lactate was completed as described by Taylor [22].

Determination of β-oxidized ketone bodies: A total of 200 mg of liver and muscle tissue were added to 1.5 mL of 0.9% NaCl and homogenized. After transferring to centrifuge tubes, 0.5 mL of PBS (1/15 M, pH 7.4) was added to each. To the sample, 0.5 mL of n-butyric acid was added, whereas 0.5 mL of water was added to the control. The determination of acetone content was performed by the indirect iodometry method [23].

Determination of CK (Creatine kinase) activity: A total of 100 mg of liver and muscle tissue was placed in a centrifuge tube containing 1 mL of sterilized saline lysis buffer and homogenized. The cells were lysed on ice for 30 min, then centrifuged at 12,000 rpm at 4 °C for 5 min. The supernatant was transferred to a 1.5-mL centrifuge tube, and the CK assay was completed as described by Lee [24].

Determination of ATP content and LDH activity: A total of 100 mg of liver and muscle tissues were added to a centrifuge tube containing 1.5 mL of sterilized saline lysis buffer, homogenized, and mixed in a centrifuge tube with 900 µL of normal saline. For the determination of LDH activity and ATP content, the procedure was completed according to the kit manufacturer’s instructions (Nanjing Jiancheng Bioengineering Institute #A020-1, A070-2, China, (Nanjing, China)).

Oxidation index determination: A total of 100 mg of liver and muscle tissue were placed in a centrifuge tube filled with 900 µL of normal saline, homogenized, centrifuged at 2500 rpm for 10 min, and the supernatant was collected. The specific procedure for the determination of catalase (CAT) activity, superoxide dismutase (SOD) activity, and malondialdehyde (MDA) concentration was completed according to the kit manufacturer’s instructions (Nanjing Jiancheng Bioengineering Institute #A007, A001-3, A003-1, China).

### 2.4. Analysis of Gene Transcriptional Activity

RNA was extracted from tissue samples (50 mg) from normal and hibernation induction stages using Trizol. DNA was isolated by the classical method. RNA and DNA from each tissue were extracted independently three times and the concentration of RNA and DNA was measured on a NanoDrop2000. The OD260/OD280 was in the normal range of 1.8–2.0, and the integrity of the RNA was assessed on a 1.5% agarose gel.

### 2.5. Analysis of mRNA Expression of Histones and (De)methylated Genes

To study the expression of histones and (de)methylated genes in various *P. sinensis* tissues at room temperature and under hibernation induction, real-time quantitative PCR experiments were performed using *P. sinensis GAPDH* as an internal reference gene [25]. Sequences for the histone genes *H1*, *H2A*, *H2B*, *H3*, and *H4*; histone methylation genes *ASH2L*, *KMT2A*, and *KMT2E;* and the histone demethylation genes *KDM1A*, *KDM1B*, and *KDM5A* were obtained from the NCBI database. Primers (Table 1) were designed using Primer5.0 software [26] and synthesized by Sangon Bioengineering (Shanghai) Co. Ltd. (Shanghai, China).

The M-MuLV first-strand cDNA synthesis kit (Sangon Bioengineering (Shanghai) Co. Ltd., #B532435, China) was used for reverse transcription, and the cDNA was stored at −20 °C for future use. The cDNA quality was verified by ordinary PCR using *GAPDH* primers. The amplification reaction (10.8 µL) included 5 µL Tap Master Mix (Dye) buffer (Kangwei Century #CW0682, (Ningbo, China)), 0.4 µL upstream and downstream primers, 4 µL ddH_2_O, and 1 µL cDNA. Amplification of the target sequence was performed according to the following protocol: One cycle for 5 min at 94 °C, followed by 32 cycles of denaturation at 94 °C for 30 s, annealing at 57.5 °C for 30 s, extension at 72 °C for 30 s, and 1 final cycle at 72 °C for 10 min. The PCR products were evaluated by 1.5% agarose gel electrophoresis, photographed with a BIO-BAD gel imager, and stored.

Using cDNA from the two groups as a template, the TB Green^®^ Premix Ex Tap II (Tli RNaseH Plus) kit (TaKaRa#RR820Q, (Kyoto, Japan)) was used according to the manual and quantitative PCR (RT-PCR) was completed using a Light Cycler 480 instrument. The results were quantitated and statistically analyzed using the 2^−ΔΔCt^ method [27].

### 2.6. Analysis of Total HMT and DM Activity

Enzyme activity assays were performed using fish total histone methylation (HMT) ELISA (enzyme-linked immune assay) and fish demethylation (DM) ELISA kits (Shanghai Enzyme Link Biotechnology Co., Ltd., #m1201940, m1239034, (Shanghai, China)) as follows: For sample processing, 50 mg of tissue sample was added to a centrifuge tube containing 1 mL of 1× PBS, pH 7.4, homogenized, and centrifuged. Then, 10 μL of the supernatant was added to the precoated enzyme-labeled plate, and the assay was completed following the manufacturer’s instructions. A microplate was coated with purified fish total histone methylation antibody. Then, the samples were sequentially added to the HRP-labeled monoclonal antibody-coated microwells. The total histone methylation antibody was combined to form an antibody-antigen-enzyme-labeled antibody complex. After thorough washing, TMB substrate was added for color development. TMB is converted to a blue color by the HRP enzyme and, finally, to a yellow color after the addition of acid. The color intensity is positively correlated with the total histone methylation in the sample.

### 2.7. Analysis of ASH2L and KDM5A Protein Expression

Testicular tissue was collected in June, September, and December from sexually mature turtles (*n* = 5 per month), which were normally reared without artificial intervention at temperature regulation. Slaughtering and sampling methods remain the same as before. Testis tissue was thoroughly milled and lysed in 1 mL of RIPA buffer (50mM Tris (pH 7.4), 150mM NaCl, 1% Triton X-100, 1% sodium deoxycholate, 0.1% SDS) on ice for 30 min, centrifuged at 10,000× *g* at 4 °C for 20 min, and the supernatants were collected. The total protein was detected using β-actin as a control, and rabbit polyclonal anti-ASH2L (1:1500, D121917, Sango, (Shanghai, China)) antibody and KDM5A (1:1500, D262113, Sango) antibody were used for immunodetection. Electrophoresis was performed by 8% SDS-PAGE, and after protein transfer and antibody incubation, the signals were visualized using an ECL substrate (NcmECL High, P2100, New Cell & Molecular Biotech Co., Ltd. (Shanghai, China)). The relative density of the protein bands was quantified using ImageJ 1.51 software.

Immunohistochemistry was performed on 4% paraformaldehyde (PFA) fixed testis sections (6 μm). Immunodetection was performed using rabbit polyclonal specific anti-ASH2L (1:300, D121917, Sango, (Shanghai, China)) antibody and KDM5A (1:300, D262113, Sango, (Shanghai, China)) antibody, stained according to the manufacturer’s instructions, mounted, and observed by microscopy.

### 2.8. Statistical Analysis

The results were analyzed by *t*-test with SPSS Version 24.0 software (SPSS, Chicago, IL, USA) [28]. Each experiment was repeated at least three times independently. All data were expressed as the mean ± standard deviation (*n* = 3). *p* values < 0.05 were considered statistically significant, (*) *p* < 0.05 was significant, and (**) *p* < 0.01 was extremely significant.

## 3. Results

### 3.1. Establishment of a Hibernation Induction Model

*P. sinensis* was exposed to a hibernation induction environment artificially. The physiological parameters are shown in Table 2. For room temperature and oral temperature less than 15 °C, a half-minute heartbeat less than 10 times, and blood oxygen concentration less than 88%, along with other indicators, indicated a hibernation or cold torpor state. As the room and water temperature gradually declined, the temperature of the carapace decreased. The number of internal strokes of the hind legs over 20 s and the length of the neck extension concomitantly decreased with the depth of hibernation induction. Additionally, the time required to turn over gradually increased. Based on these results (Table 2 is marked in blue), an analysis by one-way ANOVA indicated that there was a significant difference between the normal and the hibernation induction states (*p* < 0.05).

### 3.2. Measurement of Physiological Indices

The results of Figure 1a–f show that the glycogen content in the liver tissue was the highest, and the difference was significantly higher compared with that in muscle (*p* < 0.05). This indicates that glycogen is primarily stored in the liver, and although the temperature decreases, glycogen as an energy supply remains the same as that during a normal period. However, during hibernating and normal periods, the lactic acid content in the liver and muscle did not change significantly (*p* > 0.05), which indicates that anaerobic respiration primarily occurs in muscle tissue during hibernation induction. The acetone content increased significantly during the hibernation induction period (*p* < 0.05), indicating that there is a greater use of fat for energy metabolism during hibernation induction. CK enzyme activity in muscle was higher under normal compared with hibernating conditions (*p* < 0.01). There was no significant difference in enzyme activity in the Liver (*p* > 0.05), and ATP concentration in muscle tissue decreased significantly (*p* < 0.01); however, there was no significant difference in the liver (*p* > 0.05). LDH activity in the liver and muscle decreased significantly (*p* < 0.05), indicating that these organs exhibit less energy metabolism. During hibernation induction, CAT and SOD activity and MDA levels in the liver and muscle, decreased significantly (*p* < 0.05) (Figure 1g–i). The results indicate that oxidative stress, based on MDA as an index, was not strong in the soft-shelled turtle.

### 3.3. Analysis of Transcriptional Activity

RNA and DNA were extracted from the heart, liver, spleen, lung, kidney, ovary, adipose, muscle, small intestine, and testis of *P. sinensis* during normal and hibernating periods, and transcriptional activity was determined by the ratio of RNA to DNA. The results showed that there were significant differences in these values among different tissues of *P. sinensis* at different time periods. Further analysis showed that the RNA/DNA ratios of normal tissues were significantly higher compared with that of hibernating tissues (Figure 2, *p* < 0.01).

### 3.4. Analysis of Histone Gene Expression

The relative expression of histone gene mRNA in different tissues of *P. sinensis* was determined by qRT-PCR. As shown in Figure 3a–e, there were significant differences in histone gene expression at different stages in *P. sinensis* (*p* < 0.05). Histone *H1*, *H2A*, *H2B*, *H3,* and *H4* were expressed in all tissues; however, expression in the normal group was significantly higher compared with that in the hibernating group (*p* < 0.05).

### 3.5. Analysis of HMT and DM mRNA Expression

The relative expression of HMT and DM mRNA in different tissues of *P. sinensis* was measured by qRT-PCR. The mRNA expression of histone methylation and demethylation genes are shown in Figure 4a–e. The histone methylation genes *ASH2L*, *KMT2A,* and *KMT2E* were expressed in all tissues. The relative expression levels in the normal group were lower compared with that in the hibernation induction group. The histone demethylation genes *KDM1A*, *KDM1B,* and *KDM5A* were also expressed in all tissues and were opposite to that of the histone methylation genes.

### 3.6. Determination of Total HMT and DM Activity

The activity of histone methylation and demethylation in all tissues was significantly different changed (*p* < 0.05) (Figure 5a,b). The histone methylation activity increased (*p* < 0.05), while the histone demethylation activity decreased from the normal period to the hibernating period (*p* < 0.05).

### 3.7. Expression of ASH2L and KDM5A

The expression of the ASH2L and KDM5A proteins in the testis of *P. sinensis* during different months was measured by Western blot analysis (Figure 6a,b). In the testis, the expression of ASH2L protein increased from June to September but decreased significantly in December (*p* < 0.05); whereas the expression of KDM5A increased significantly from June to December (*p* < 0.05).

The distribution of the ASH2L and KDM5A proteins in the testis of healthy *P. sinensis* during different months was determined by immunohistochemistry (Figure 7a–i). The nuclei were stained blue with hematoxylin and brown–yellow by DAB (3,3′-Diaminobenzidine tetrahydrochloride, AR1000, Boster Biological Technology Co., Ltd. (Wuhan, China)). The results indicated that the ASH2L and KDM5A proteins were expressed in the testis during all seasons, but there were slight differences among seasons, especially in September and December. The main difference was the proportion of cells in the testis during each season. As shown in Figure 7, both ASH2L and KDM5A were expressed in testis tissue, mainly in the nucleus and cytoplasm. Positive expression was also observed in spermatogonia (Spg) and deformed sperm cells (Dsc), whereas negative expression was found in spermatocytes (S) and mature sperm (Ms). The negative control tissues showed no positive reactions, such as brown particles.

## 4. Discussion

Hibernation is a common phenomenon in Animalia. The temperature of hibernating animals is below 15 °C in the environment and in the mouth, the heart rate drops to 10–20 beats per minute, and blood oxygen concentration decreases [2,29]. These results are consistent with the decrease in room temperature and heart rate in the present study, which indicated that *P. sinensis* was in a hibernation state and the model was established successfully. The study of physiological metabolism reflects specific changes in hibernation. Energy metabolism indicators, such as glycogen and lactate, have not changed in the liver and muscle during hibernation, indicating that glycogen is mainly stored in the liver as an energy-supplying substance to provide energy. Meanwhile, the acetone concentration does increase. As the muscle is the tissue with the most lactic acid, which is produced by glycogen fermentation, it has lower oxygen content and more frequent anaerobic respiration during hibernation. In addition, acetone is a product of fatty acid metabolism, suggesting that a hibernating fatty acid metabolism is more active in the liver and muscles, which is consistent with T. G. West’s study of hibernating frogs [30]. The decrease in exercise during hibernation in *P. sinensis* resulted in a sharp decrease in energy expenditure and a significant decrease in the total CK enzyme activity and ATP concentration in hibernating muscle, but there was no significant difference in the liver before and after hibernation. Furthermore, the total CK activity and ATP content of the heart muscle of *Chrysemys picta bellii* were relatively stable during long-term hypoxia [5,6,31]. During hibernation, the activity of LDH in the liver and muscle of *P. sinensis* decreased significantly [32]. This may result from its high antioxidant capacity, which remains stable during hibernation; however, the activity of CAT and SOD and MDA levels in the liver and muscle decreased significantly during hibernation. MDA is a marker of oxidative stress injury, which indicates that there was less oxidative stress in the liver and muscle of *P. sinensis* during hibernation, or the antioxidative stress of *P. sinensis* itself is stronger [32,33]. This is consistent with the results of a cold stress recovery study by Chen [34], in which a reduction in SOD activity was observed. However, in the cold exposure study of Prawn [33], SOD and MDA increased initially and then decreased, which was probably due to a difference in species. In the present study, the blood oxygen concentration, the number of heartbeats, and the oral temperature of *P. sinensis* all decreased with a decrease in room temperature. We determined that the Chinese soft-shelled turtle entered the hibernation state as the energy metabolism and antioxidant indices decreased.

Hibernating animals use hypothermia to suppress transcription and reduce metabolic expenditure [35]. Studies have shown that during hibernation, the body’s metabolism and transcriptional activity are completely suppressed [7,12,36]. DNA content was not affected by cell processes at different stages, but RNA content varied with the rate of protein synthesis [37], which was consistent with the results of this experiment. Histones are closely associated with and perform an important role in gene transcription and translation [38]. In this study, histone gene expression analysis revealed a higher level in the normal environment compared with the hibernating environment, which was in direct proportion to the transcription level [39]. These results suggest that histones perform an inhibitory role in transcriptional regulation during hibernation induction in *P. sinensis*. Similarly, histones can control the transcription rate of an organism through histone modification and regulation. Previous studies have shown that histone lysine methylation (*H3K4me*) is closely associated with gene activation and transcriptional regulation [40], which control key aspects of normal cell physiology and development. For example, during embryonic stem cell differentiation in mice, the knockout of *ASH2L* resulted in transcriptional suppression and downregulation of mitotic-related genes [41,42]. Studies have also shown that the knockout of *MLL1/5* also decreases histone methylation levels and inhibits chromatin remodeling and cell growth [43,44]. These results are consistent with the present study, indicating that *H3K4* methylation is actively regulated, which may perform a role in regulating the transcriptional activation of genes or promote the growth and development of tissues and the protection of organisms under low temperature and hypoxia. In contrast, the histone demethylation genes *LSD1*/2 and *KDM5A* are responsible for the demethylation of *H3K4*, resulting in transcriptional inactivation and transcription inhibition [45]. Knockout of *LSD1* also promotes the differentiation of embryonic stem cells, so *LSD1*-mediated decreased expression of *KDM5A* may be beneficial for cell development and differentiation [45,46,47,48]. *KDM5A* has a transcriptional inhibitory effect by removing the *H3K4me3* mark on active gene promoters [49]. In the present study, histone demethylation enzymes were significantly reduced during hibernation, and the overall level of *H3K4* demethylation was inhibited [50,51]. These results indicate that it may inhibit the transcription of *P. sinensis* during hibernation induction, protect the organism in an extreme environment, and regulate the growth and development of *P. sinensis*.

The maturation process of sperm is governed by unique transcriptional regulation and numerous chromatin alterations; however, histone methylation is one mechanism that regulates gene expression at the transcriptional level. Currently, a large amount of evidence indicates that histone methylation modification is closely related to spermatogenesis in mammals and the induction of germ cell loss by aberrant epigenomes in male germ cells [52,53]. In human sperm, histones near certain developmental genes are always retained in a methylated form [54]. During mammalian spermatogenesis, H3K4 methylation is distributed in both space and time, and the protein level of lysine-specific histone demethylase LDS1 (AOX2) in testis tissue is higher compared with that in somatic tissue. These results are consistent with the qRT-PCR results. The histone methylation and demethylation genes are mainly located in spermatogonia [55] and round and elongated spermatids [56], which is why we chose testis tissue for Western blot analysis and immunohistochemistry. ASH2L was previously shown to perform a role in embryonic morphogenesis and open chromatin [42]. In this study, ASH2L protein levels exhibited a significant downward trend during hibernation, indicating that histone methylation status did not perform a role in activating transcription in extreme environments, which is consistent with ASH2L working together with other molecules to promote transcriptional activation [57]. Moreover, ASH2L is a nuclear protein that is rarely expressed in the cytoplasm. During the process of spermatogenesis, the cytoplasm is lost, and the nucleus accounts for the majority, so the entire spermatogenic cell is positively stained. The deletion of ASH2L results in the downregulation of many mitotic-related genes [41], which is closely associated with the transformation of sperm morphology. In previous studies, KDM5A (RBP2), as a transcriptional repressor, can inhibit transcription by demethylating H3K4 in its promoter, which is consistent with the decrease in transcription observed during hibernation and the increase in KDM5A protein levels, which regulate transcriptional repression. The KDM5A protein is localized in the nuclei of mouse germ cells but only weakly expressed in type A and B spermatogonia [40]. Immunohistochemistry showed that KDM5A is localized to the nuclei of gonadal cells, spermatogonia, and spermatocytes [58]. However, the positive expression of spermatogonia and deformed spermatids in this study was somewhat different from that of previous studies, which may be the result of species differences. The results of this experiment suggest that histone (de)methylated genes are closely associated with spermatogenesis and that testis tissue may synthesize and translate histone (de)methylated proteins. The coexistence of histone methylation and demethylation in the same cell suggests that they work in concert to regulate germ cell development and localization in spermatogonia, suggests an important role for this gene, which undergoes self-renewal to ensure a persistent supply of germ cells to form sperm. The mRNA expression of ASH2L and KDM5A combined was the highest in the testes, and the protein was localized in each germ cell. We believe that these two genes are closely associated with spermatogenesis in the Chinese soft-shelled turtle.

## 5. Conclusions

Our results are the first to demonstrate that the oxidation markers CAT, SOD, and MDA of *P. sinensis* decrease during hibernation induction. The metabolic indices of total CK, ATP, and LDH associated with energy consumption also showed a downward trend, and the metabolic indices of glycogen related to energy supply also decreased. Acetone exhibited an increasing trend, whereas lactate, which is a metabolic indicator related to anaerobic respiration, also showed an increasing trend. Thus, artificial hibernation induction was successfully established in *P. sinensis* for the first time. As a result, the whole transcriptional activity of *P. sinensis* during hibernation induction was decreased, and the relative expression of histone proteins was also significantly decreased. This indicates that the transcription of *P. sinensis* is inhibited when hibernating genes are inactive. The relative expression of histone methylation and histone methylation enzyme activity were significantly increased, and the relative expression of demethylation genes and demethylation enzyme activity were significantly decreased. Western blot analysis and immunohistochemistry indicated that histone *H3K4* methylation performs an important role in mitosis, chromatin remodeling, and cell growth during hibernation. It is also necessary for normal cell development and differentiation; however, future proteomics and enzyme functional studies will be necessary to develop a more comprehensive understanding of hibernation induction.

## Figures and Tables

**Figure 1 biology-12-00720-f001:**
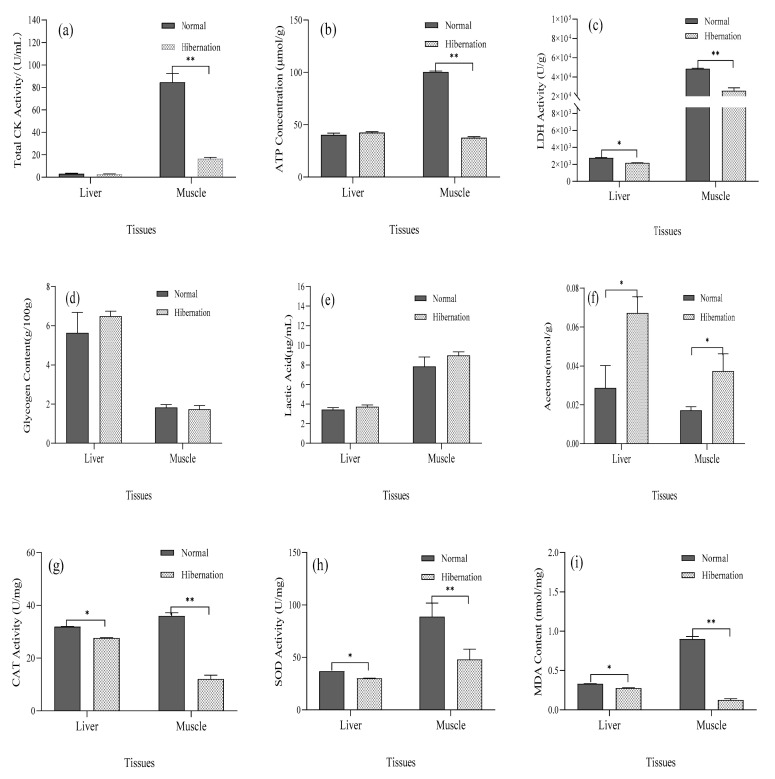
Determination of physiological indices in *P. sinensis* liver and muscle at different stages. (**a**) Total CK activity of muscle and liver during normal and hibernation induction periods, (**b**) ATP levels in muscle and liver during normal and hibernation induction periods, (**c**) LDH activity in muscle and liver during normal and hibernation induction periods, (**d**) Glycogen content in muscle and liver during normal and hibernation induction periods, (**e**) Lactic acid content in muscle and liver during normal and hibernation induction periods, (**f**) Acetone levels in muscle and liver during normal and hibernation induction periods, (**g**) CAT activity in muscle and liver during normal and hibernation induction periods, (**h**) SOD activity in muscle and liver during normal and hibernation induction periods, and (**i**) MDA content in muscle and liver during normal and hibernation induction periods. (*n* = 6; *: *p* < 0.05, **: *p* < 0.01).

**Figure 2 biology-12-00720-f002:**
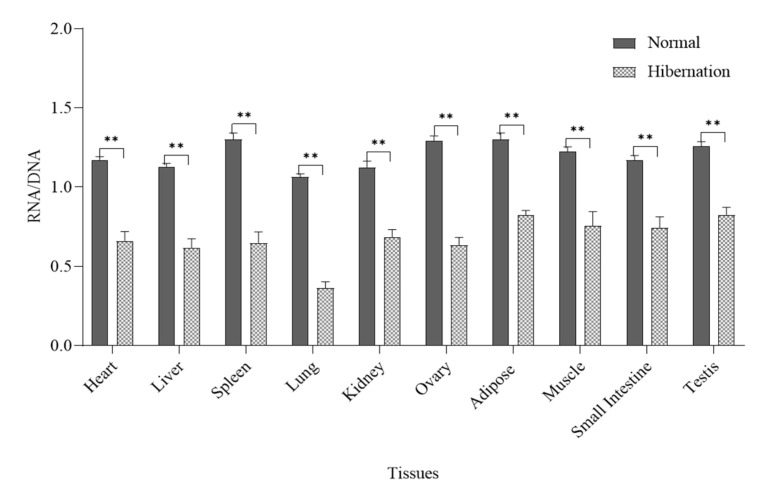
The ratio of RNA/DNA in various tissues of *P. sinensis* during normal and hibernation induction periods. (Ovary and testis, *n* = 3, others, *n* = 6; **: *p* < 0.01).

**Figure 3 biology-12-00720-f003:**
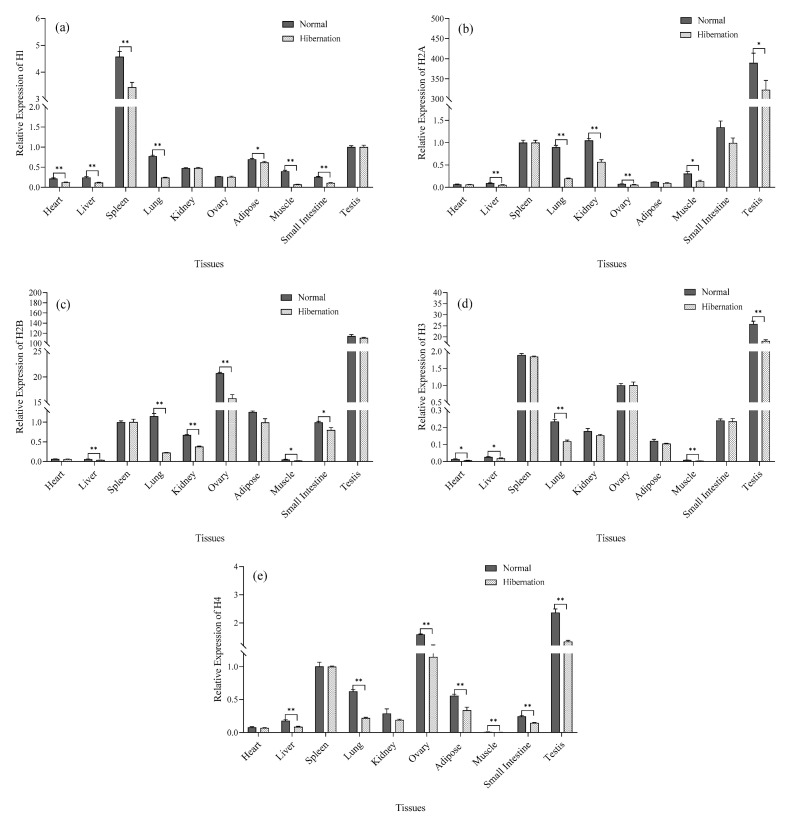
Relative expression levels of histone mRNA in various tissues of *P. sinensis* during normal and hibernation induction periods. (**a**–**e**) Relative expression of histones *H1*, *H2A*, *H2B*, *H3,* and *H4*. (Ovary and testis, *n* = 3, others, *n* = 6; *: *p* < 0.05, **: *p* < 0.01).

**Figure 4 biology-12-00720-f004:**
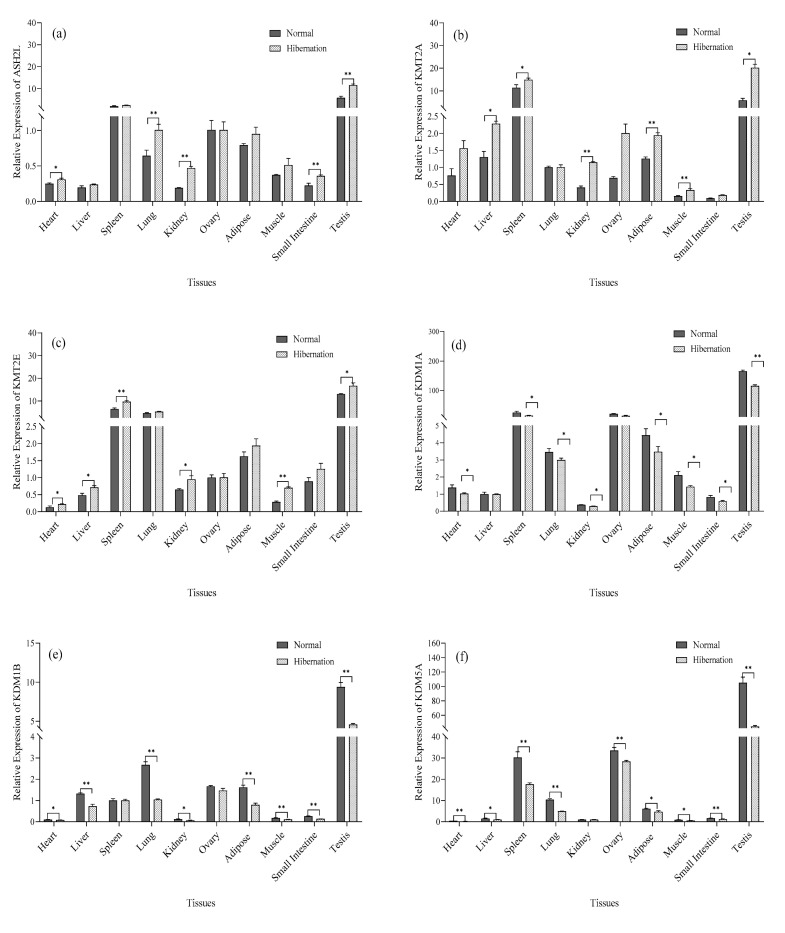
mRNA expression levels of HMT and DM mRNA in various tissues of *P. sinensis* during normal and hibernation induction periods. (**a**–**c**) Relative expression of the histone methylation genes *ASH2L*, *KMT2A,* and *KMT2E*, (**d**–**f**) Relative expression levels of the histone demethylation genes *KDM1A*, *KDM1B,* and *KDM5A*. (Ovary and testis, *n* = 3, others, *n* = 6; *: *p* < 0.05, **: *p* < 0.01).

**Figure 5 biology-12-00720-f005:**
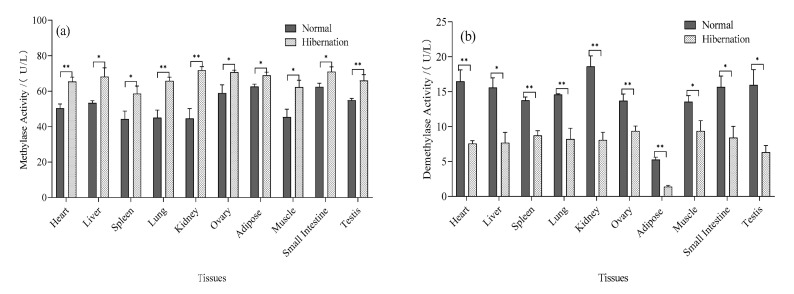
HMT and DM enzyme activities in various tissues of *P. sinensis* at different stages. (**a**) HMT activity in different tissues of *P. sinensis* during normal and hibernation induction periods, (**b**) DM activity in different tissues of *P. sinensis* during normal and hibernation induction periods. (Ovary and testis, *n* = 3, others, *n* = 6; *: *p* < 0.05, **: *p* < 0.01).

**Figure 6 biology-12-00720-f006:**
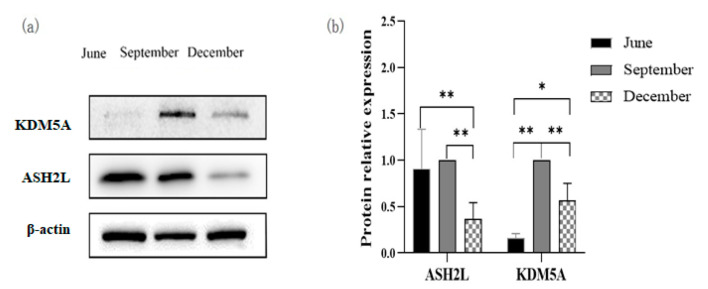
Relative expression of ASH2L and KDM5A proteins in the testis of *P. sinensis* during different months. (**a**) The protein expression of ASH2L and KDM5A by Western blot analysis, (**b**) Relative quantitative analysis of ASH2L and KDM5A protein expression in the testis by WB. (*n* = 5 per month; *: *p* < 0.05, **: *p* <0.01) (Appendix A: Full western blot of ASH2L and KDM5A.).

**Figure 7 biology-12-00720-f007:**
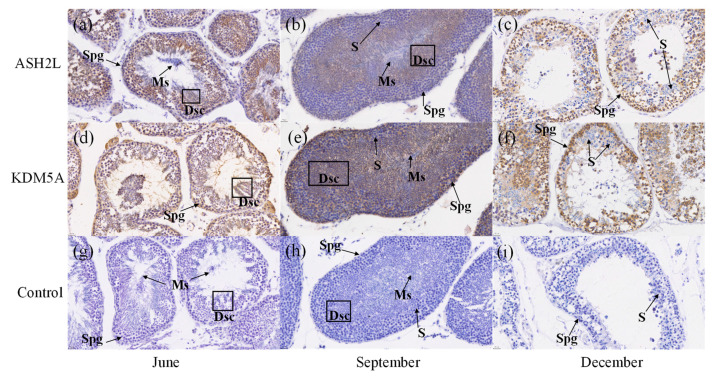
Localization of ASH2L and KDM5A proteins in the testis of Trionyx sinensis in different months. (**a**–**c**) show the localization of ASH2L protein in the testis during different months, (**b**,**d**–**f**) show the localization of KDM5A protein in the testis during different months, (**c**,**g**–**i**) show the blank control for testis during different months. (Scale bar = 20 μm). Spg, spermatogonia; S, spermatocytes; Dsc, deformed sperm cells; Ms, mature sperm.

**Table 1 biology-12-00720-t001:** Forward and reverse primers used for quantitative PCR.

Target Gene	Primers	Primer Sequence(5′-3′)	Login ID
*GAPDH*	*GAPDH*-F *GAPDH*-R	CCTGGTATGACAATGAGTT GTGCCTGGTTTATTCCTT	NM_001286927.1
*H1*	*H1*-F *H1*-R	TCCTGCTGTGTCCGCTCCTG GGAAGACTTACGGGCTTTGGAACC	XM_006125889.3
*H2A*	*H2A*-F *H2A*-R	AAGGTCAGTGGAAACTCTGGTTGC TGCCTTACTTGCTGGTCTGTGTTC	XM_006136395.2
*H2B*	*H2B*-F *H2B*-R	GAGGGTCGGTCGAGATGTCTACG GACTCCTTGCGGCTCTTCTTACG	XM_025177927.1
*H3*	*H3*-F *H3*-R	GAAATCGCCCAGGACTTCAAGACC GGCATGATGGTGACTCGCTTAGC	XM_006125911.2
*H4*	*H4*-F *H4*-R	CTAAGGTGCCTTGAGTCTGCTGTC CGAGCCAAGCGACGAATAGCC	XM_006125900.3
*ASH2L*	*ASH2L*-F *ASH2L*-R	ACTGACCGTTATTGGCGAGAAAGG CAAGTCTGGCTGCTGTGTCTGG	XM_006134041.3
*KMT2A*	*KMT2A*-F *KMT2A*-R	TGCGATTCCGACACTTGAAGAAGAC TGAGGATGGAGCGAATGACATTGC	XM_006120948.3
*KMT2E*	*KMT2E*-F *KMT2E*-R	AGTGTGGTAAGGCTGCTTGTAAGTG TGGTGAGGAGGATCAGGCTTCTATC	XM_014580823.2
*KDM1A*	*KDM1A-F* *KDM1A-R*	AACGAAGAAGACTGGCAAGGTGATC CTTCCAGAAGCGTGACATCCATCC	XM_014568580.2
*KDM1B*	*KDM1B-F* *KDM1B-R*	GAAGCAGCAGAGGATGATGATGAGG TAGCACACCTTTCAGCAGCACTTG	XM_006121822.3
*KDM5A*	*KDM5A-F* *KDM5A-R*	CAAGGCTACAGGTGTGGTCTCAAG GCTGCTGATTGTAGGCTGGTATCC	XM_014579673.1

**Table 2 biology-12-00720-t002:** Changes in physiological parameters before and after hibernation.

RT (°C)	WT (°C)	CT (°C)	PT (°C)	OT (°C)	HHM	BOC (%)	TOT (s)	LNE (cm)
26	20.1	20.6 ± 0.17	21.3 ± 0.83	18.7 ± 0.90	13.0 ± 0.82	93.3 ± 2.36	3.3 ± 0.47	15.0 ± 1.63
23	18.9	19.6 ± 0.21	21.4 ± 0.41	18.3 ± 0.73	11 ± 0.82	95 ± 0.00	6.3 ± 0.94	15 ± 0.82
20	16.1	18.0 ± 0.21	19.4 ± 0.50	17.5 ± 0.47	12.3 ± 2.62	95 ± 0.00	5.7 ± 0.94	16 ± 0.82
17	15.2	18.0 ± 0.21	19.4 ± 0.50	17.5 ± 0.47	12.3 ± 2.62	91.7 ± 2.36	6.7 ± 1.25	14.7 ± 1.25
14	14.3	15.2 ± 0.25	16.2 ± 0.33	16.7 ± 0.87	17.0 ± 5.10	88.3 ± 2.36	6.0 ± 0.82	14.7 ± 1.25
11	10.9	13.1 ± 0.29	13.2 ± 0.54	14.5 ± 0.24	6.7 ± 1.25	86.7 ± 2.36	7.3 ± 2.36	14.7 ± 1.25
8 *	8.4 *	9.6 ± 0.21 **	8.9 ± 0.47 *	13.1 ± 0.34 *	5.0 ± 0.82	81.7 ± 2.36 **	9.0 ± 2.45	8.0 ± 0.00

Note: RT = Room Temperature; WT = Water Temperature; CT = Carapace Temperature; PT = Plastron Temperature; OT = Oral Temperature; HHM = Heartbeats in Half a Minute; BOC = Blood Oxygen Concentration; TOT = Turn Over Time; LNE = Length of Neck Extension; * = Significant differences between values at normal and hibernation induction; ** = Significant polar differences between normal and hibernation induction (*n* = 10; *: *p* < 0.05; **: *p* < 0.01).

## Data Availability

The raw data presented in this study are available on request from the corresponding author.

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
