# Peer review of "Induction of Hibernation and Changes in Physiological and Metabolic Indices in *Pelodiscus sinensis"

_biology, 2023, doi:10.3390/biology12050720_

Round 1
Reviewer 1 Report (Previous Reviewer 1)
This manuscript has been revised accroding to my suggestions well.
Author Response
Thank you for your recognition of our work.
Reviewer 2 Report (Previous Reviewer 3)
In most cases, freshwater turtles will not really hibernate at a temperature of >5 degree Celsius in subtropical region. The inactivity of turtles in cold environments is the physiological effects of hypothermia rather than hibernation. The authors have to define and discuss clearly in the introduction. If turtles have not really entered a period of dormancy, it would be better to use "cold torpor" or "overwintering".
Author Response
Response: We agree to your suggestion. For the state after cooling induced hibernation, cold torpor might be a more accurate description in this experiment. We have modified it as “cold torpor” or "hibernation induction" throughout the article.
The description added in Simple Summary as follows:” Although this paper lacks further evidence that P. sinensis has been induced into a deep hibernation state, cold torpor is perhaps a more accurate description.”
The description added in Abstract as follows:” Although our results showed significant changes in physiological and gene expression after hibernation induction, we could not confirm that the P. sinensis entered deep hibernation. Therefore, for the state after cooling induced hibernation, cold torpor might be a more accurate description. ”

This manuscript is a resubmission of an earlier submission. The following is a list of the peer review reports and author responses from that submission.
Round 1
Reviewer 1 Report
This manuscript entitled “ Induction of Hibernation and Changes in Physiological and Metabolic Indices in Pelodiscus Sinensis” is an interesting research, it is helpful to understand the mechanism of hibernating for Pelodiscus Sinensis. There are some suggestions which would improve further the quality of the manuscript.
1. Simple summary: The topic of this study is about hibernation, and the objective is to study the changes of physiological metabolism index, histones, and related genes on growth and development under conditions of artificially induced hibernation. However, it is described “…… studied the effects of the physiological metabolism index, histones, and related genes on growth and development under conditions of artificially induced hibernation.” I suggested replacing “effects” with “changes” in this sentence.
2. The objective of this manuscript is about the regulatory mechanism of hibernation, why there are results about the expression of ASH2L and KDM5A proteins in the testis of P. sinensis during different months? Why western blot analysis was not used under hibernation.
3. The western blot analysis and ELISA were employed in this study, but the antibody is not from P. sinensis, how to illustrate its effectiveness.
4. Which is the sample number? In Table 2, n=50, it is not correct.
5. The hibernation sample is conducted on which temperature?
6. In this study, there are two groups data need to analyze, the statistical analysis should be T-test, instead of One-way ANOVA.
7. Table 1, the primers of KDM1A, KDM1B and KDM5A were missing.
Reviewer 2 Report
In the paper by Lin et al. the authors have studied how hibernation in the Chinese freshwater turtle Pelodiscus Sinensis affects metabolism, histone expression, and expression and activity of histone methylases and demethylases. The authors found that hibernation lowered the activity of creatine kinase in muscle and of lactate dehydrogenase in liver and muscle. Interestingly, the glycogen and lactic acid content in liver and muscle did not change, while acetone concentration went up. CAT and SOD activities, and MDA concentration was lower during hibernation. During hibernation, histone gene expression was lowered, and while histone methylase expression and activity increased, the expression and activity of demethylases decreased. It is well known that gene expression is regulated by histone methylation and demethylation, and the authors demonstrate that this mechanism is also used in P. Sinensis to regulate gene expression during hibernation.
The manuscript is overall written in a comprehensible manner, and the finding could be of interest. However, I have some serious concerns regarding this paper.
Major concerns:
In nature, P. Sinensis hibernate at the bottom of ponds and thus under hypoxic conditions. I would have expected to see a decrease in the glycogen content and an increase in lactic acid, but in this study, there were no changes in liver and muscle (Figure 1), although the heart rate and blood oxygen concentration was reduced (Table 2). It is even more puzzling that the authors in the first paragraph of the Discussion (page 12) state that “Energy metabolism indicators, such as glycogen, lactate, and acetone, were increased during hibernation”. The acetone concentration does increase, but as noted above, the glycogen and lactic acid concentrations do not change with hibernation according to Figure 1d and e.
It is not clear for me, why the authors chose to focus on the testis to study the expression of ASH2L and KDM5A through Western blotting and immunohistochemistry (Figures 6 and 7). Why not study this in other organs, whose metabolism would probably be more important during hibernation?
In the experiments with artificial hibernation, you show that the ASH2L expression increased, whereas the KDM5A expression decreased during hibernation (Figure 4 a and f). However, when you assessed the expression in turtles collected at different times of the year (Figure 6), the turtles sampled in December, which is when the turtles normally hibernate, had lower ASH2L and higher KDM5A expression. These results seem contradictory.
Furthermore, it is not clear where the animals from these experiments (in figures 6 and 7) came from, as they were taken at different times of the year. I assume that they were at different from the animals that were artificially hibernated, but this is not mentioned in the Materials and methods.
Overall, the n-numbers are not clear for any of the experiments. The n-number of animals of each sex for each experiment should be stated in every table and figure legend. In section 2.1 of the Materials and methods, it is stated that 50 turtles, half male and half female, were obtained for the experiments. But in section 2.2, it is stated that three of the animals were selected as parallel samples and sacrificed for the experiments. What happened with the rest of the turtles? If they died during the experiment, then this should be reported, but raises serious concerns about the method of artificial hibernation.
I did not see any mention that this study has been approved by and ethics committee.
It is stated that tissue was homogenized in RIPA buffer for determination of CK and LDH activity. The exact composition of the RIPA buffer is not given, but RIPA buffer usually contains SDS and sodium deoxycholate, which both are denaturing enzymes. Thus, it would not be possible to record any enzyme activity in these homogenates.
In section 3.5, you state that “The expression changed regularly with the period and in different periods for different tissues of P. Sinensis (p < 0.05)”. I do not understand this sentence, as there was only normal and hibernation state.
Minor concerns:
Could you please state the ratio of dark:light and whether it was changed during the artificial hibernation.
I was unable to find the full names of CAT (catalase), SOD (superoxide dismutase), and MDA (malondialdehyde) in the text. Their function and importance should have been introduced in the text.
Section 3.2: You state that “There was no significant difference in enzyme activity in the Liver (p < 0.05)” and later – regarding the ATP concentration – that “there was no significant difference in the liver (p < 0.05)”
Section 3.5: I think you want to state that the mRNA expression is shown in Figures 4a-f.
Legend of Figure 4: There are only subpanels a-f in this figure, yet in the legend, you mention subpanels i-k.
Section 3.6, second sentence: You mention demethylation activity twice here, but I think you mean to first say talk about methylation and then demethylation.
Section 3.7: I think you mean to refer to Figure 6a-b instead of 5a-b.
Figure 6: Thank you for showing a larger picture of your KDM5A blot than the usual crop just around the bands. But can you please indicate which band you analysed.
Reviewer 3 Report
1. Hibernation of turtles is a behavioral and physiological response to extremely low temperature, usually <5 C. The authors have to clarify the differences between hibernation and overwintering. In most cases, the activity levels of turtles will lower sharply as temperature decreases, but they will not enter dormancy (hibernation) at a temperature of 8 C (it must be much higher in core body temperature).
2. Oral (buccal) temperature is not a valid method to assess core body temperature for ectothermic reptiles. It is also inappropriate to measure the surface temperature of carapace and plastron by using an IR thermometer in wet condition.
3. The experiments have to be well designed to verify the turtles have really entered the phenomenon of dormancy. In the studies of ectothermic reptiles, cloacal temperature has been widely used as a valid measurement of body temperature rather than buccal temperature.